# Regulation of IDO2 by the Aryl Hydrocarbon Receptor (AhR) in Breast Cancer

**DOI:** 10.3390/cells12101433

**Published:** 2023-05-20

**Authors:** Sarah Y. Kado, Keith Bein, Alejandro R. Castaneda, Arshia A. Pouraryan, Nicole Garrity, Yasuhiro Ishihara, Andrea Rossi, Thomas Haarmann-Stemmann, Colleen A. Sweeney, Christoph F. A. Vogel

**Affiliations:** 1Center for Health and the Environment, University of California, One Shields Avenue, Davis, CA 95616, USA; sykado@ucdavis.edu (S.Y.K.); kjbein@ucdavis.edu (K.B.); castaneda@ucdavis.edu (A.R.C.); aapouraryan@ucdavis.edu (A.A.P.); nlgarrity@ucdavis.edu (N.G.); 2Graduate School of Integrated Arts and Sciences, Hiroshima University, Hiroshima 739-8521, Japan; ishiyasu@hiroshima-u-ac.jp; 3Leibniz Research Institute for Environmental Medicine, 40225 Düsseldorf, Germany; andrea.rossi@iuf-duesseldorf.de (A.R.); thomas.haarmann-stemmann@iuf-duesseldorf.de (T.H.-S.); 4Department of Biochemistry & Molecular Medicine, School of Medicine, University of California, Davis, CA 95817, USA; casweeney@ucdavis.edu; 5Department of Environmental Toxicology, University of California, One Shields Avenue, Davis, CA 95616, USA

**Keywords:** AhR, breast cancer, IDO, IDO2, immunity, TCDD, PM, tumor microenvironment

## Abstract

Indoleamine 2,3-dioxygenase 2 (IDO2) is a tryptophan-catabolizing enzyme and a homolog of IDO1 with a distinct expression pattern compared with IDO1. In dendritic cells (DCs), IDO activity and the resulting changes in tryptophan level regulate T-cell differentiation and promote immune tolerance. Recent studies indicate that IDO2 exerts an additional, non-enzymatic function and pro-inflammatory activity, which may play an important role in diseases such as autoimmunity and cancer. Here, we investigated the impact of aryl hydrocarbon receptor (AhR) activation by endogenous compounds and environmental pollutants on the expression of IDO2. Treatment with AhR ligands induced IDO2 in MCF-7 wildtype cells but not in CRISPR-cas9 AhR-knockout MCF-7 cells. Promoter analysis with IDO2 reporter constructs revealed that the AhR-dependent induction of IDO2 involves a short-tandem repeat containing four core sequences of a xenobiotic response element (XRE) upstream of the start site of the human *ido2* gene. The analysis of breast cancer datasets revealed that IDO2 expression increased in breast cancer compared with normal samples. Our findings suggest that the AhR-mediated expression of IDO2 in breast cancer could contribute to a pro-tumorigenic microenvironment in breast cancer.

## 1. Introduction

Breast cancer is the most common cancer among women worldwide, with an estimated 2.3 million new cases per year as of 2020, accounting for nearly one in eight of all cancers diagnosed according to the World Health Organization [1]. Despite advances in diagnostic and treatment modalities, breast cancer presents a significant clinical challenge, with approximately 685,000 women succumbing to the disease annually [1]. This is, in part, due to tumor heterogeneity, inherent differences in breast cancer subtypes, and the degree of therapeutic response [2,3]. It is therefore essential to characterize the pathologic signaling pathways involved in breast neoplasia to identify novel molecular targets that may serve as a basis for the development of future breast cancer therapies. The aryl hydrocarbon receptor (AhR) is an emerging signaling pathway that is garnering considerable promise as a therapeutic target for certain subtypes of breast cancer [4,5].

The AhR is activated by a diverse range of ligands, including the endogenous compounds kynurenine (Kyn) and 6-formylindolo [3,2-b]carbazole (FICZ), and natural compounds such as dietary flavonoids, as well as exogenous compounds such as the prototypical AhR ligand 2,3,7,8-tetrachlorodibenzo-p-dioxin (TCDD), polycyclic aromatic hydrocarbons (PAHs), and polychlorinated biphenyls (PCBs) [6,7,8]. Moreover, environmental pollutants such as ambient air particulate matter (PM) derived from traffic and urban areas have also been found to contain compounds that activate the AhR and increase the risk to develop malignant diseases [9,10,11]. Ligand binding induces conformational changes in the AhR that promote nuclear translocation, heterodimerization with the aryl hydrocarbon receptor nuclear translocator (ARNT), and the transcriptional regulation of downstream targets in a xenobiotic response element (XRE)-dependent manner [7]. The downstream targets of AhR include the induction of the enzymes involved in xenobiotic metabolism and detoxification, mainly cytochrome P450 (CYP)1A1 and CYP1B1, as well as immune regulatory enzymes such as IDO1 and IDO2 [7,12]. Aside from AhR’s critical role in physiological processes, the AhR has also been implicated in pathological processes such as inflammation and cancer [13,14,15]. The role that the AhR plays in pathological processes is an active area of study; however, the effects mediated by the AhR appear to be dependent on cell type, ligand specificity, and the microenvironment [16,17,18]. In the context of cancer, the AhR has been shown to promote cell migration, angiogenesis, and tumor proliferation [19,20]. Conversely, some studies have provided evidence for a tumor suppressor role for AhR, underscoring the complexity of the AhR pathway [21,22]. 

Furthermore, the activation of AhR in DCs promotes immune suppression via induction of regulatory T-cell (Treg) differentiation. Tregs in turn inhibit the function of the effector T cells dedicated to detecting and eliminating cancer cells, effectively bypassing immune surveillance [23]. In this context, the AhR is activated by Kyn metabolites, which are tryptophan byproducts produced by the immune regulatory enzyme tryptophan 2,3-dioxygenases (TDO) and the closely related enzymes indoleamine 2,3-dioxygenase 1 (IDO1) and indoleamine 2,3-dioxygenase 2 (IDO2). IDO1 and IDO2 are therefore considered important players in the suppression of T-effector cells and other immune cells, creating a global environment that promotes tumorigenesis. 

In an initial study, Munn et al. [24] uncovered the critical role of IDO1 as an immunosuppressive enzyme to prevent allogeneic fetal rejection. About 9 years later, IDO2, also called indoleamine 2,3-dioxygenase-like-protein 1 (INDOL1), was discovered by Ball et al. [25] and has been described in both mice and humans. Physiologically, both IDO1 and IDO2 are considered critical enzymes in immune regulation and tolerance via the metabolism of tryptophan in the kynurenine pathway [26,27]. Pathologically, IDO1 can be induced by Interferon (IFN)γ where it counteracts inflammation by promoting immune tolerance through T-cell anergy, regulatory T-cell (Treg) differentiation, and the inhibition of natural killer (NK) cell cytotoxicity [26,27,28]. Although useful for counteracting excess inflammatory responses and collateral tissue damage, in the context of cancer, excess IDO activity dampens cancer immunosurveillance, as supported by recent studies indicating that IDO facilitates tumorigenesis and cancer immune evasion [29]. These findings are further supported by studies demonstrating Treg cell induction and the increased expression of programmed death ligand-1 (PDL-1) as essential steps in immune suppression in various types of cancers, including breast cancer [27,29]. While the regulation of IDO1 by IFNγ and pathogens has been reported, the induction of IDO2 by pro-inflammatory cytokines and inflammatory stimuli is controversial. Interestingly, the expression of IDO1 and IDO2 was induced by IFNγ in various human cancer cell lines [30], which is in contrast with a study showing no effect of IFNγ on IDO2 expression [25,27]. In patients with breast cancer, IDO expression has been reported to be upregulated in tumor cells and associated with poor prognosis and resistance to therapy [31,32]. Therefore, IDO1 and IDO2 may serve as promising immunotherapeutic targets in breast cancer, as IDO inhibition can enhance antitumor immunity and thereby restore cancer immunosurveillance. In fact, clinical trials in cancer patients using vaccines against IDO and PDL-1 have shown promising results with prolonged survival [33]. An important role of IDO in tumor development was also supported by the finding that the expression of IDO by cancer cells correlated with poor clinical prognosis in ovarian carcinoma [34] and endometrial carcinoma [35]. Furthermore, an increased expression of IDO1 was found in triple-negative breast cancers and was associated with PD-L1 co-expression, suggesting that IDO1 might be a mechanism of anti-PD-1/PD-L1 immunotherapy resistance [33]. Several studies investigated the expression of IDO1 in tumor tissue and cancer cells; however, the role and the expression of IDO2 are not well studied in breast cancer. A recent study found the expression of *IDO1* and *IDO2* in many cancer types, including breast, colon, and endometrial cancers [36].

The expression of IDO1 and IDO2 differs in specific types of tumors [30]. Targeting the AhR–IDO2 axis in breast cancer could be a promising therapeutic approach to enhance antitumor immunity and improve clinical outcomes. Indeed, IDO2 upregulation has been observed in numerous cancers, including non-small cell lung cancer, pancreatic cancer, colon cancer, gastric cancer, and renal tumors [37]. Interestingly, the reduced tumor growth of lung carcinoma cells was found in IDO2-knockout mice compared with wildtype mice [38]. Moreover, IDO2 absence in this setting altered the tumor microenvironment and increased the accumulation of tryptophan, resulting in an enhanced immune cell infiltration, which indicates that IDO2 may play a critical role in shaping the tumor microenvironment. A higher mRNA expression of IDO2 was also found in adipocyte stem cells, which are important stromal cells in the tumor microenvironment, isolated from breast cancer patients [39]. Interestingly, a recent study in a zebrafish model found that the treatment of adipocyte stem cells with the potent AhR agonist TCDD led to increased stemness and metastasis of co-cultured breast cancer cells [40]. Elucidating the precise relationship between the AhR and IDO2 may shed light on novel therapeutic strategies for future breast cancer treatments. The focus of this study is to investigate the regulation of IDO2 expression in human breast cancer by AhR ligands, including environmental pollutants and endogenous ligands known to activate the AhR signaling pathway.

## 2. Materials and Methods

### 2.1. Reagents and PM Preparation

Dimethyl sulfoxide (DMSO) was obtained from Sigma. Interferon γ (IFNγ) was purchased from R&D Systems (Minneapolis, MN, USA). Benzo(a)pyrene (BaP) was provided by Sigma-Aldrich (St. Louis, MO, USA). TCDD (>99% purity) was originally obtained from Dow Chemical Co., Ltd. (Midland, MI, USA). Other molecular biological reagents were purchased from Qiagen (Valencia, CA) and Roche Clinical Laboratories (Indianapolis, IN, USA). Kynurenine (Kyn) and 6-formylindolo [3,2-b]carbazole (FICZ) were purchased from Cayman Chemicals (Ann Arbor, MI, USA). The stock solutions of AhR ligands were dissolved in DMSO and control cells received 0.1% DMSO. The traffic-related air pollution (TRAP)-related PM_2.5_ was collected from an exposure facility located at a major freeway tunnel system near San Francisco, California, as described in [41]. PM samples were collected using an impaction-based filter system and chemically extracted according to Bein and Wexler [42,43].

### 2.2. Generation of CRISPR/Cas9 AhR Mutants of MCF-7 Cells

The generation of CRISPR/Cas9 AhR mutants of MCF-7 cells has been described recently [44]. In brief, the CRISPR design tool CHOPCHOP (http://chopchop.cbu.uib.no/ (accessed on 21 March 2023)) was used to design gRNA targeting AhR exon 2 (5′-AAGTCGGTCTCTATGCCGCTTGG-3′). The gRNA was cloned into a PX458 plasmid (Addgene 48138). MCF-7 cells were transfected with nuclease plasmids in an antibiotic-free medium using FuGENE HD (Roche), according to the manufacturer’s protocol. Cells were sorted (FACS) and plated as single cells in a 96-well plate after 48 h. Clones were genotyped using high-resolution melt analysis and SANGER sequencing. AhR knockout was also confirmed using a DNA/RNA Shield™ kit (Zymo Research, Irvine, CA, USA).

### 2.3. Cell Culture and Transfection Experiments

The epithelial breast cancer cell line MCF-7 wildtype (wt) *and AhR knockout* were maintained in DMEM. The cell culture medium contained 10% fetal bovine serum and 100 units of penicillin and 100 µg/mL streptomycin. Cells were seeded in 24-well plates and transfected using jetPrime (PolyTransfection; Qbiogene, Irvine, CA, USA), according to the manufacturer’s protocol. The IDO2 promoter constructs were suspended in 50 µL of JetPrime reagent. The JetPrime solution was incubated at room temperature for 10 min. The transfection was allowed to proceed for 16 h, and the cells were treated with various ligands of the AhR for 24 h. The luciferase reporter construct containing the IDO2 promoter sequences was provided by SwitchGear Genomics (Menlo Park, CA, USA) corresponding to a −3275 bp of the human IDO2 promoter sequence. The 432 bp IDO2 promoter construct (Δ-2563-2131) containing the STP with four putative AhR binding sites was created by Thomas Haarmann-Stemmann. The IDO2 luciferase reporter constructs were amplified and purified with a Zymo PURE-Endo Zero plasmid isolation kit (Zymo Research, Irvine, CA, USA).

### 2.4. RNA Isolation and Real-Time PCR

Total RNA was isolated from cells using a Quick-RNA Mini prep isolation kit (Zymo Research), and cDNA synthesis was performed as described in [45], using a cDNA synthesis kit by Applied Biosystems (Foster City, CA, USA). The detection of β-actin and differentially expressed target genes was performed with a LightCycler LC480 Instrument (Roche Diagnostics, Indianapolis, IN, USA) using a Fast SYBR Green Master Mix (Applied Biosystems), according to the manufacturer’s instructions. The primers for each gene were designed on the basis of the respective cDNA or mRNA sequences using the OLIGO primer analysis software provided by Steve Rozen and the Whitehead Institute/Massachusetts Institute of Technology Center for Genome Research so that the targets were 90–250 bp in length (Table 1). PCR amplification was carried out as described in [45]. To confirm the amplification specificity, the PCR products were subjected to melting curve analysis.

### 2.5. Chromatin Immunoprecipitation (ChIP) Assay

ChIP assay was performed as previously described [45]. Briefly, MCF-7 cells were seeded in 150 mm dishes and cultured in DMEM containing 10% FBS. AhR ligands were added for the indicated times, and protein–DNA complexes were cross-linked with 1% formaldehyde for 10 min. The ChIP assays with AhR-specific antibodies were analyzed with PCR using primer pairs covering the STR region containing four XRE core sequences (two XREs in forward and two XREs in reverse–complement orientation) of the human IDO2 promoter. DNA was purified using a DNA purification kit (Zymo Research) and eluted in 50 µL. ChIP DNA (5 µL) was amplified using real-time PCR with primers 5′-GCACTATGGGAGGCTGAAG-3′ and 5′-GGCGCGATCTTGGCTCACTG-3′ covering the STR with four XREs of IDO2 to amplify a 235 bp fragment of the IDO2 promoter.

### 2.6. Statistical Analysis

The experiments in this study were repeated three times, and data are expressed as mean ± S.D. Differences were considered significant at *p* < 0.05. A comparison of two groups was performed with an unpaired, two-tailed Student’s *t*-test. A comparison of multiple groups was performed with an analysis of variance followed by Dunnett’s or Tukey’s test.

### 2.7. Analysis of Breast Cancer Datasets

The TIMER2.0 (http://timer.cistrome.org/ (accessed on 21 March 2023)) and Breast Cancer Gene-Expression Miner v4.9 (http://bcgenex.ico.unicancer.fr/BC-GEM/GEM-Accueil.php?js=1) tools were accessed on 14 May 2023 and used to assess IDO2 mRNA expression in cancer datasets, as specified in the figure legends. Statistical analyses are embedded features of these tools. The TIMER2.0 tool uses the Wilcoxon test to assess statistical significance, while the Breast Cancer Gene-Expression Miner v4.9 uses Welch’s *t*-test. When the Welch *p*-value was significant, and there were at least three different groups, the Dunnett–Tukey–Kramer’s test was used for two-by-two comparisons. 

## 3. Results

### 3.1. Induction of IDO2 and IDO1 by AhR Ligands

First, we tested the time-dependent effects of TCDD on the expression of IDO2 in MCF-7 cells. TCDD caused a time-dependent induction with the greatest increase in IDO2 after 24 h treatment with TCDD. The activation of the AhR led to a significant increase in IDO2 mRNA level at 12 h with the highest induction of eight-fold at 24 h, which was sustained over 48 h (Figure 1). TCDD did not change the expression of IDO2 in AhR-knockout MCF-7 cells. Next, we tested the effects of various AhR ligands, including TCDD, BaP, FICZ, and Kyn, on IDO2 expression (Figure 1). Recent studies have shown that the AhR may act as a sensor of chemical components in environmental pollutants such as ambient air particulate matter (PM) modifying the immune response [9]. Therefore, we included the PM_2.5_ derived from a traffic-related source (TRAP PM). TCDD (1 nM), BaP (2.5 uM), FICZ (100 nM), and TRAP PM (10 ug/mL) significantly increased the expression of IDO2 after 24 h treatment (Figure 2A). The concentrations of AhR ligands and TRAP PM were chosen based on the different binding affinities of the specific AhR ligands. The high-affinity AhR ligand TCDD at 1 nM led to a 7.8-fold increase, followed by a 6.2-fold increase with 2.5 μM BaP, and a 5.4-fold increase in IDO2 above control after treatment with 10 ug/mL TRAP PM (Figure 2A). FICZ (100 nM) induced an increase in IDO2 of about 2.5-fold, and 50–10 μM Kyn caused only a minor increase in IDO2 mRNA in MCF-7 cells, which was statistically not significant. None of the AhR ligands tested had a significant effect on the expression of IDO2 in AhR-knockout MCF-7 cells. Additionally, we tested the expression of IDO1 in comparison to IDO2 after treatment with the AhR ligands (Figure 2B). TCDD caused the highest increase in IDO1 (5.4-fold) above control, followed by PM (3.3-fold) and BaP (2.8-fold). In general, treatment with the AhR ligands led to a smaller-fold increase in the induction of IDO1 compared with IDO2. We also stimulated cells with IFNγ, which is known to induce IDO1 via the JAK–STAT–IRF1 pathway [46]. Interferon regulatory factor 1 (IRF-1) is a transcriptional activator or repressor that binds to an interferon-stimulated response element (ISRE) and regulates the expression of target genes containing functional ISREs on their promoter region. IFNγ stimulation of MCF-7 cells significantly upregulated IDO1 about 245-fold above control, but IFNγ had no effect on the expression of IDO2 at a concentration of 100 U/mL IFNγ (data not shown). This might be explained by the fact that the IDO1 promoter contains the consensus site for IRF1 binding, while the IDO2 gene contains a putative ISRE, which has not yet been shown to be functional (Figure 3A) (https://motifmap.ics.uci.edu/ accessed on 21 March 2023).

### 3.2. IDO2 Promoter Activity Is Regulated by AhR

We identified a short-tandem repeat (STR) at −2478 bp upstream from the start site of the human *ido2* gene containing four putative XRE consensus sequences, with two XREs in forward and two XREs in reverse–complement orientation (Figure 3A). Additionally, two XRE core sequences located at −1949 bp and −1400 bp and one putative interferon-stimulated response element (ISRE) at −1333 bp upstream of the start site were identified. TCDD, BaP, and PM significantly induced the promoter activity of the full-length IDO2 promoter construct in transfected MCF-7 WT cells (Figure 3B). Treatment with FICZ led only to a small increase in the IDO2 promoter activity, whereas TCDD, BaP, and PM clearly increased IDO2 promoter activity by about 2.5-fold compared with control cells. Treatment with TCDD, BaP, or PM of MCF-7 AhR ko cells transfected with the full-length IDO2 construct did not activate the IDO2 promoter. The stimulation of MCF-7 wt or AhR ko cells with IFNγ at 100 U/mL had no effect on IDO2 expression. 

### 3.3. TCDD-Induced IDO-2 Promoter Activity Is XRE-Dependent

Next, we investigated whether the XREs of the STR or the XREs downstream were important to mediate the induction of IDO2. We generated a −2325 bp deletion construct that does not contain the STR of the IDO2 promoter. TCDD did not upregulate the promoter activity of the −2325 bp deletion construct in MCF-7 WT cells as compared to the full-length (3275 bp) reporter construct (Figure 3C). We also transfected MCF-7 WT cells with a 432 bp IDO2 promoter construct (Δ-2563-2131), which contained the STR and the four putative AhR binding sites. As for the full-length construct, TCDD significantly enhanced the promoter activity of the 432 bp IDO2 promoter construct.

### 3.4. Enhanced Recruitment of AhR to the STR of the IDO2 Promoter

A ChIP assay with MCF-7 WT was performed to study the recruitment of the AhR to the STR of the IDO2 promoter. The ChIP samples were analyzed using qPCR, and the level of AhR enrichment was analyzed (Figure 4). The DNA binding activity of the AhR to the STR with its four XRE core sequences of the IDO2 promoter increased in TCDD-treated cells by about six-fold above the control level. Stimulation with IFNγ did not stimulate the recruitment of the AhR to STR.

### 3.5. Expression of IDO2 in Cancer

Using the TIMER2.0 resource [47], the expression of IDO2 across TCGA tumors was examined, as shown in Figure 5, with normal samples included as available. In general, IDO2 mRNA expression increased in cancer types relative to normal samples, although several exceptions were observed, including cholangiocarcinoma (CHOL), liver hepatocellular carcinoma (LIHC), and thyroid carcinoma (THCA). In breast cancer TCGA samples (Figure 5), IDO2 expression increased relative to normal samples (*p* = 9.40 × 10^−6^). This was also observed in the combined TCGA and GTEx datasets (Figure 6A), evaluated using the bc-GenExMiner 4.5 resource [48]. Using bc-GenExMiner 4.5, IDO2 expression was assessed in the PAM50 molecular subtypes of breast cancer, in a combined dataset of 4387 samples (TCGA, SCAN-B/GSE96058, and SCAN-B/GSE81538). As shown in Figure 6B, basal-like breast cancer samples had the highest IDO2 expression relative to other subtypes. There were no significant differences in IDO2 expression between tumor stages (data not shown). Breast cancer samples with a high proliferation index, as determined via Ki67 IHC staining, had significantly higher IDO2 expression than those with a low proliferation index (Figure 7). Collectively, these results support a pro-tumorigenic role for IDO2, but further studies are necessary. 

## 4. Discussion

The AhR is an important regulator of T-cell differentiation and immune responses [49]. As immune regulatory enzymes, IDO1 and IDO2 have been found to be important targets of AhR to mediate immunosuppression and promote the growth of cancer cells [50]. Previously, we described the TCDD-induced and AhR-dependent expression of IDO1 and IDO2 in dendritic cells derived from the human monocytic U937 cell line [12] and mouse bone-marrow-derived cells [51]. Many subsequent studies confirmed the induction of IDO1 by AhR ligands in various tissues and disease models based on its implication in the regulation of T-cell differentiation and immunosuppression (e.g., [52]). In normal tissues, IDO1 is expressed by endothelial cells in the placenta, lung, colon, lymph nodes, testis, epididymis, thyroid, and spleen, as well as by epithelial cells in the female genital tract [53]. A different pattern of expression has been described for IDO2 compared with IDO1 [54,55,56]. A constitutive level of IDO2 has been found in the liver, kidney, brain, lymph nodes, placenta, and endometrium [25,53,54,55,56]. An increased expression of both IDO1 and IDO2 has been observed in various types of cancer [30,34,35] including an AhR-mediated expression of IDO2 in the human mammary epithelial cell line MCF10A [57]. In this study, the analysis of publicly available gene expression datasets revealed that IDO2 increased in various cancer types, including breast cancer, suggesting a pro-tumorigenic role for IDO2. However, further functional studies will be needed.

While the enzymatic function of IDO1 to metabolize tryptophan and induce the differentiation of T regulatory cells by increasing levels of Kyn has been well defined, the role of IDO2 is less clear. Compared with IDO1, IDO2 has a relatively low level of enzymatic activity [58,59]. While IDO2 may collectively contribute to the metabolism of Tryp, IDO2 also involves non-enzymatic and pro-inflammatory functions. Recent studies reported that IDO2 may play an important role in diseases such as autoimmunity and cancer [60]. Therefore, both IDO1 and IDO2 may play an important role in cancer by inducing tumor-immune tolerance through enzyme-dependent and enzyme-independent IDO-mediated immunosuppression [61,62,63]. However, the relevance of IDO2 in the suppression of the anticancer immune response and tumor promotion is still unclear [64].

In the current study, we found that toxic AhR ligands such as TCDD, BaP, and TRAP-derived PM significantly induce the expression of IDO2 in MCF-7 human breast cancer cells in an AhR-dependent manner. However, the endogenous AhR ligands FICZ and Kyn had only a moderate or no effect on IDO2 expression. These results confirm our earlier study in DCs showing a robust increase in IDO2 mRNA by TCDD but only a minor induction of IDO2 by FICZ [12]. A promoter study with the full-length construct and a deletion construct of the IDO2 promoter identified four XRE core sequences located on the STR responsible for the AhR-induced expression of IDO2 after treatment with TCDD, BaP, or PM. A putative RelBAhRE and two other XRE core sequences seem to play a minor role in the upregulation of IDO2 by AhR ligands. As for IDO2 mRNA expression, Kyn and FICZ had only a moderate or no effect on IDO2 promoter activity. Our promoter study revealed that an XRE-dependent and canonical AhR signaling pathway is responsible for the upregulation of IDO2 expression. The results also indicate a ligand-specific induction of IDO2, considering the weak effects of Kyn and FICZ compared with TCDD, BaP, and PM. In contrast to IDO1, which is regulated and induced by IFNγ, the expression of IDO2 was not affected by IFNγ at 100 U/mL, confirming a previous study with U937-derived DCs [12]. However, it is possible that the effect of IFNγ on IDO2 is cell-type-specific or that higher concentrations of IFNγ are required to induce IDO2, as shown in various human cancer cell lines [30]. It is also noted that the IDO1 promoter contains an IRF1 consensus site permitting induction via the JAK–STAT–IRF1 pathway. The results of the gene expression and promoter studies indicate that IDO2 is uniquely regulated by the AhR, whereas IDO1 is regulated by both the AhR and IFNγ. It is interesting to note that IDO1 and IDO2 not only regulate T cells but may also modulate the function of B cells [65,66]. Even though IDO2 is closely related to IDO1 and encoded by a linked gene via gene duplication [67], the IDO isozymes can differ in their biological functions, as elegantly shown in an in vivo model of autoimmune arthritis revealing IDO2′s function as a pro-inflammatory mediator of B- and T-cell immune responses [68]. This is important since the AhR has been found to play an important role as a mediator linking inflammation and breast cancer [69]. The studies suggest different mechanisms regarding how IDO2 and IDO1 may contribute to pathological outcomes. Consequently, it is critical to elucidate the detailed mechanisms and distinguish the individual contribution of IDO1 vs. IDO2 in cancer and/or autoimmune diseases. More work is needed to elucidate IDO2′s role in AhR-mediated pathways that can be targeted therapeutically in chronic diseases.

## Figures and Tables

**Figure 1 cells-12-01433-f001:**
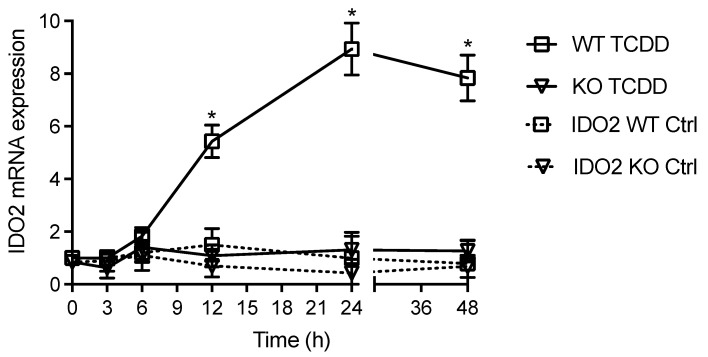
Time-dependent expression of IDO2 after treatment with TCDD in wildtype and AhR-knockout MCF-7 cells. Cells were treated with 1 nM TCDD for various time points for 3, 6, 12, 24, and 48 h and then harvested for RNA extraction and IDO2 expression analysis via qPCR after 24 h treatment. TCDD stock solution was dissolved in DMSO and control cells received 0.1% DMSO. The expression was corrected against the housekeeping gene ß-actin. Results are presented as mean ± SEM, and the *y*-*axis* represents mRNA expression level. The expression level of IDO2 mRNA in control cells is shown in dotted lines; * significantly higher than the control (*p* < 0.05).

**Figure 2 cells-12-01433-f002:**
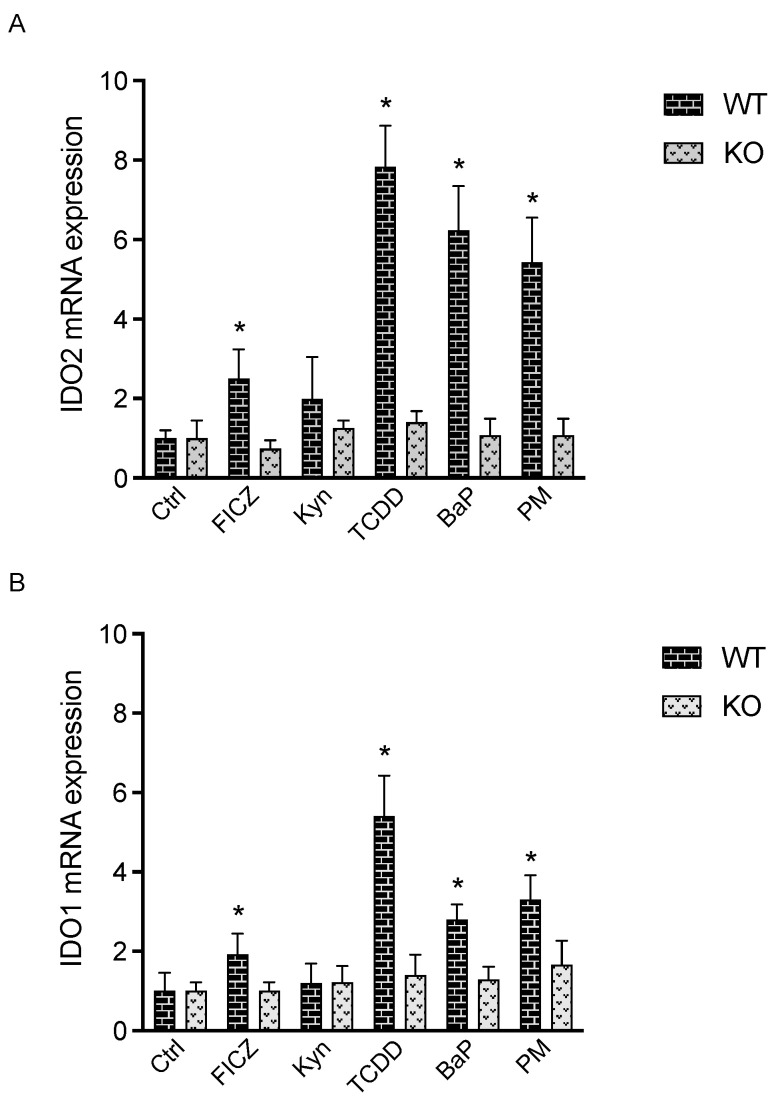
Effect of AhR ligands on the expression of IDO1 and IDO2 in MCF-7 cells. MCF-7 cells were treated with FICZ (100 nM), Kyn (50 µM), TCDD (1 nM), BaP (2.5 µM), and TRAP PM (10 µg/mL) for 24 h. In addition, cells were treated with 100 U/mL IFNγ (data not shown), which induced a 245-fold increase in IDO1 but had no effect on IDO2 expression. Cells were harvested for RNA extraction and (**A**) IDO2 and (**B**) IDO1 expression analysis via qPCR after 24 h treatment. The expression was corrected against the housekeeping gene ß-actin. Results are presented as mean ± SD, and the *y*-*axis* represents the mRNA expression level; * significantly higher than control (*p* < 0.05).

**Figure 3 cells-12-01433-f003:**
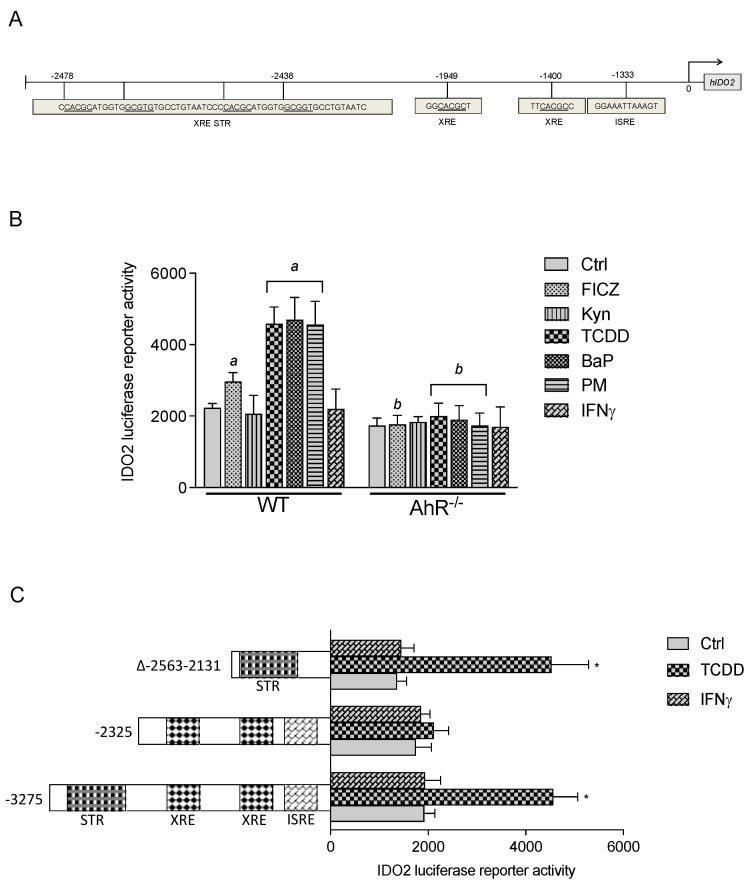
TCDD-induced IDO2 promoter activity is AhR- and XRE-dependent: (**A**) Schematic illustration of the promoter construct of the mouse *ido2* gene containing 3275 bp upstream of the transcriptional start site cloned into a luciferase (*luc*) reporter vector. The positions of the short-tandem repeat (STR) at −2478 bp containing four putative XRE consensus sequences, two XRE consensus sites at −1949 bp and −1400 bp, and one recognition site for ISRE at −1333 bp are shown. (**B**) Wildtype (WT) and AhR-knockout (AhR^−/−^) MCF-7 cells were transfected with the IDO2 luciferase reporter construct containing 3275 bp of the human *ido2* gene promoter region; ^*a*^, significantly higher than WT control cells (*p* < 0.05); *^b^*, significantly lower than treated WT cells. (**C**) MCF-7 WT cells were transfected with the full-length (3275 bp) IDO2 reporter plasmid, a −2325 bp deletion construct, and a 432 bp (Δ−2563-2131) construct containing the STR sequence with four XRE core elements. Cells were transfected for 16 h and treated with 1 nM TCDD or 100 U/mL IFNγ for 6 h. Relative luciferase activity units are given as mean values of triplicates as a result of three independent experiments; *, significantly different from control cells (*p* < 0.05).

**Figure 4 cells-12-01433-f004:**
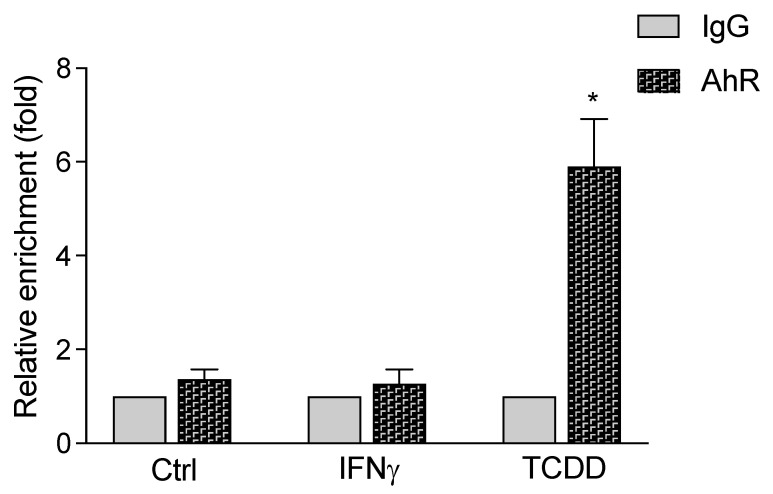
ChIP assay for STR region of the human IDO2 promoter. TCDD induces the recruitment of AhR to the STR region. MCF-7 WT cells were treated with 1 nM TCDD or 100 U/mL IFNγ, for 6 h, and anti-AhR antibodies were used for immunoprecipitation. Data show the ratio of the enrichment of the promoter region target in TCDD-treated cells. The increase in the enrichment was calculated relative to that of the negative control with anti-IgG. qPCR was performed to analyze the levels of AhR binding to the STR on the IDO2 promoter. Error bars indicate standard deviations from the mean of at least three experiments; * significantly higher than control (*p* < 0.05).

**Figure 5 cells-12-01433-f005:**
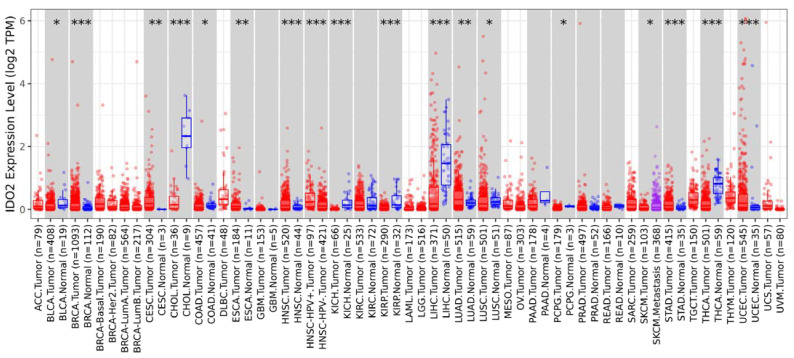
Expression of IDO2 mRNA in cancer. The TIMER2.0 tool was used to assess IDO2 mRNA expression in the TCGA dataset across cancer types using the Gene_DE module under the “Exploration” Table Tumor expression is depicted in red, and when available, normal samples are depicted in blue. Samples in purple are metastatic samples (only available in SKCM). *: *p*-value < 0.05; **: *p*-value < 0.01; ***: *p*-value < 0.001. The abbreviations for cancer types are as follows: ACC—adrenocortical carcinoma, BLCA—bladder urothelial carcinoma, BRCA—breast invasive carcinoma, CESC—cervical and endocervical cancer, CHOL—cholangiocarcinoma, COAD—colon adenocarcinoma, DLBC—diffuse large B-cell lymphoma, ESCA—esophageal carcinoma, GBM—glioblastoma multiforme, HNSC—head and neck Cancer, KICH—kidney chromophobe, KIRC—kidney renal clear cell carcinoma, KIRP—kidney renal papillary cell carcinoma, LAML—acute myeloid leukemia, LGG—low-grade glioma, LIHC—liver hepatocellular carcinoma, LUAD—lung adenocarcinoma, LUSC—lung squamous cell carcinoma, MESO—mesothelioma, OV—ovarian serous cystadenocarcinoma, PAAD—pancreatic adenocarcinoma, PCPG—pheochromocytoma and paraganglioma, PRAD—prostate adenocarcinoma, READ—rectum adenocarcinoma, SARC—sarcoma, SKCM—skin cutaneous melanoma, STAD—stomach adenocarcinoma, TGCT—testicular germ cell tumors, THCA—thyroid carcinoma, THYM—thymoma, UCEC—uterine corpus endometrial carcinoma, UCS—uterine carcinosarcoma, UVM—uveal melanoma.

**Figure 6 cells-12-01433-f006:**
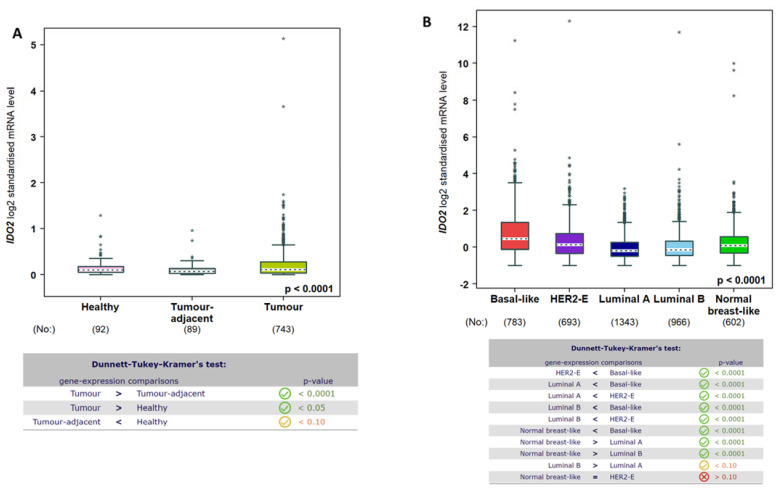
Expression of IDO2 in breast cancer: (**A**) The Breast Cancer Gene−Expression Miner v4.9 tool was used to assess IDO2 mRNA expression (RNA-seq) in the combined TCGA and GTEx datasets. (**B**) The Breast Cancer Gene−Expression Miner v4.9 tool was used to assess IDO2 mRNA expression (RNA-seq) in the combined TCGA and SCAN-B datasets as a function of PAM50 breast cancer subtypes, as indicated. For both (**A**,**B**), sample numbers are indicated in parentheses beneath the plot. Statistical analysis is an embedded feature of this tool, described in Materials and Methods. Welch *t*-test, *p*-value < 0.001.

**Figure 7 cells-12-01433-f007:**
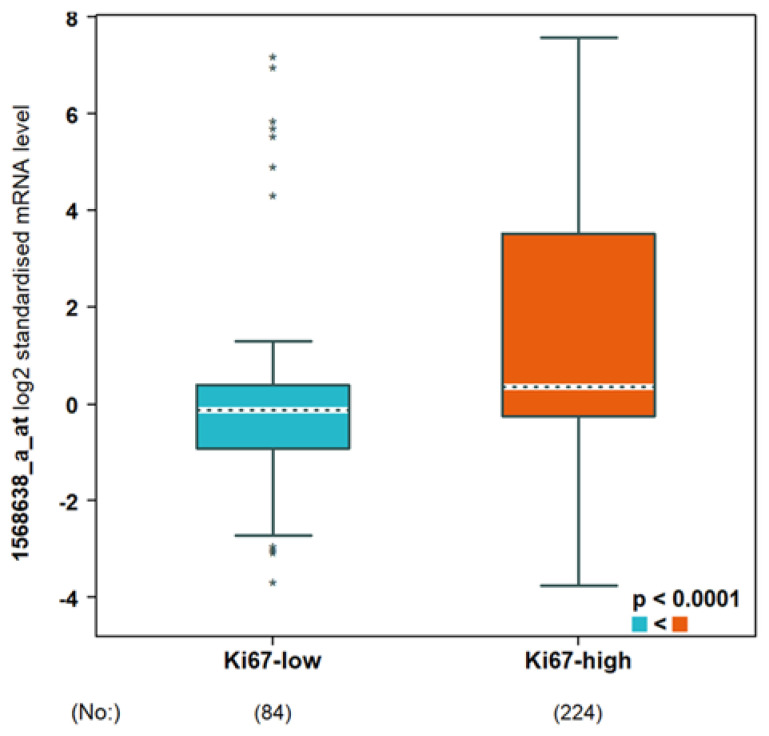
Expression of IDO2 in Ki67-low and Ki67-high breast cancers. The Breast Cancer Gene−Expression Miner v4.9 tool was used to assess IDO2 mRNA expression (1568638_a_at, Affymetrix expression data) in breast tumors that were Ki67-low vs. Ki67-high, as assessed via Ki67 immunohistochemistry. Statistical analysis is an embedded feature of this tool with the Welch *t*-test’s *p*-value included in the plot inset. Sample numbers are indicated in parentheses beneath the plot.

**Table 1 cells-12-01433-t001:** PCR primer for human target genes.

Gene	Forward Primer	Reverse Primer	PCR Product Size
β-actin	catccgcaaagacctgtacg	cctgcttgctgatccacatc	218
IDO1	caggcagatgtttagcaatga	gatgaagaagtgggctttgc	91
IDO2	cctgatcactgcttaacggc	ttggaggcagtgctcagtat	214

## Data Availability

Publicly archived datasets analyzed in this study can be found at http://bcgenex.ico.unicancer.fr/BC-GEM/GEM-Accueil.php?js=1 (accessed on 21 March 2023) and http://timer.cistrome.org/ (accessed on 21 March 2023).

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
