# Peer review of "Regulation of IDO2 by the Aryl Hydrocarbon Receptor (AhR) in Breast Cancer"

_cells, 2023, doi:10.3390/cells12101433_

Round 1

Reviewer 1 Report

The manuscript by Kano et al., Regulation of IDO2 by the AhR in breast cancer includes a set of well design experimental approaches, including effects of AhR different type of agonists , namely TCDD, BaP, FICZ and Kyn. In addition, a traffic-related air pollution (TRAP)-related PM2.5 obtained from samples collected from an exposure facility located at freeway tunnel was also included in the study.  The study is complemented with an analysis of breast cancer datasets to study IDO2 expression in human breast cancer samples. The method used seem appropriate, the experimental design correct and results solid.

However, the study has some flaws that, in my view, make it unacceptable in the present version. Below are the points that should be improved.

(i) The study aims “to investigate if environmental pollutants known to activate the AhR signaling pathway influence IDO2 expression.” 

i.i. Since a diverse range of AhR ligands were used,  some of them are endogenous compounds, the focus on environmental pollutants seems to be non appropriate.

i.ii. How the datasets analysis fit in this aim?

(ii) The Introduction section is to long and not focused on the aim (or aims) of the study. 

ii.i. What is the relevance of the interaction between AhR and estrogen signaling for the aim of the study?

ii.ii The same question for the immunomodulatory effects of AhR: what is the connection with the aim presented?

(iii) The original studies on the involvement of AhR in the effects of air pollutants should deserve greater attention. Studies on a connection between AhR, air pollution and cancer appear in the 90s of the last century. For greater accuracy in this introduction, pioneering studies should be cited for recognition..

(iv) Results

iv.i. The study was carried out in breast MCF-7 cells (luminal epithelial phenotype of breast cells) submitted or not to AhR Ko. The authors mentioned the “effects mediated by the AhR appear to be highly dependent on cell type…” line 62.  However, only a single cell line was used which has a luminal epithelial phenotype and poor metastatic capacity. Does it not represent a limitation? Please comment.

iv.ii. line 218: confirm concentration of BaP

iv.iii. Figure 1: why control cells were treated with 0.1% DMSO? Because drugs were diluted in 0.1% DMSO? Probably! But were is this stated?

iv.iv. Figure 2 should be improved. The inclusion of the IFNg effects make it hard to compare effects on IDO1 and IDO2. I suggest to remove the effects of IFNg and adjust the scale to make it simple the comparison between Fig 2A and Fig 2 B. The IFNg results can be included in the legend since making a figure to show only the effects of IFNg is useless with such difference in response (the effects on IDO2 will be to small to be seen.

iv.v. Figure 5: may not the statistical significance be presented with the p< 0.05)?

iv.v. A comparison between the different stages can not be done for the different cancer subtypes?

(v.) Discussion

The discussion section has two distinct parts with a break at line 380. After line 381 it is easy to follow the rational. It is discussed the results obtained in the present study in the MCF-7 cells. The part before line 380 is a more general discussion of previous works of the group. In my view, it would be preferable to include these information after and not before the discussion of  present data. 

Furthermore, the analysis of breast cancer datasets are not discussed. The absence of this set of results in the discussion and in the aims, make it hard to understand the link between the in vitro studies and the dataset analysis. It seems that they were “artificially” combined with a rational to be studied in parallel. 

(vi)  There is a lack of consistency in the references format. It seems the authors forgot to review this part of the manuscript.

Author Response

Reviewer 1

We thank the Reviewer for their careful analysis of our manuscript and comments which have improved our manuscript.

The manuscript by Kano et al., Regulation of IDO2 by the AhR in breast cancer includes a set of well design experimental approaches, including effects of AhR different type of agonists , namely TCDD, BaP, FICZ and Kyn. In addition, a traffic-related air pollution (TRAP)-related PM2.5 obtained from samples collected from an exposure facility located at freeway tunnel was also included in the study.  The study is complemented with an analysis of breast cancer datasets to study IDO2 expression in human breast cancer samples. The method used seem appropriate, the experimental design correct and results solid.  However, the study has some flaws that, in my view, make it unacceptable in the present version. Below are the points that should be improved.

(i) The study aims “to investigate if environmental pollutants known to activate the AhR signaling pathway influence IDO2 expression.” 

i.i. Since a diverse range of AhR ligands were used, some of them are endogenous compounds, the focus on environmental pollutants seems to be non-appropriate.

Answer: Indeed, besides environmental pollutants BaP, TCDD and PM, we also tested endogenous AhR ligands.  We have mentioned this in the abstract line 24 and Introduction line 125.

i.ii. How the datasets analysis fit in this aim?

Answer: Very little is known regarding IDO2 expression in cancer. Evaluating its expression in human tumor samples is one aspect of understanding how it may play a role as an immune regulatory enzyme in the tumor microenvironment. In the revised manuscript (Figure 5), we provide a pan-cancer evaluation of IDO2 mRNA expression and find that in most cancers, IDO2 expression is increased. We also provide a focused evaluation of its expression in breast cancer (Figures 6, 7).

(ii) The Introduction section is too long and not focused on the aim (or aims) of the study. 

ii.i. What is the relevance of the interaction between AhR and estrogen signaling for the aim of the study?

Answer: Thank you for this feedback. Parts of the Introduction have been removed including the sentence mentioning the interaction between AhR and estrogen signaling in line 63-65 and 68-69.

ii.ii The same question for the immunomodulatory effects of AhR: what is the connection with the aim presented?

Answer: The immunomodulatory effects of AhR are important for its effect on the TME which impacts tumor progression.

(iii) The original studies on the involvement of AhR in the effects of air pollutants should deserve greater attention. Studies on a connection between AhR, air pollution and cancer appear in the 90s of the last century. For greater accuracy in this introduction, pioneering studies should be cited for recognition.

Answer: Thank you for this comment. Citations of early studies showing the connection between AhR and air pollution have been added [10,11] on page 2, line 53.

(iv) Results

iv.i. The study was carried out in breast MCF-7 cells (luminal epithelial phenotype of breast cells) submitted or not to AhR Ko. The authors mentioned the “effects mediated by the AhR appear to be highly dependent on cell type…” line 62.  However, only a single cell line was used which has a luminal epithelial phenotype and poor metastatic capacity. Does it not represent a limitation? Please comment.

Answer: Thank you for this comment. Indeed, many reports show that the downstream effects of AhR activation depend on the specific cell type as mentioned in the Introduction line 62. However, we have previously shown that AhR mediates induction of IDO2 expression in macrophages, dendritic cells and MCF10A mammary epithelial cells as cited in line 59, line 363-64, and line 373. This indicates that although AhR signaling is cell context-dependent, a common outcome of AhR signaling is the induction of IDO2 expression.

iv.ii. line 218: confirm concentration of BaP

Answer: Concentration of BaP has been confirmed in line 216.

iv.iii. Figure 1: why control cells were treated with 0.1% DMSO? Because drugs were diluted in 0.1% DMSO? Probably! But where is this stated?

Answer: Stock solutions of AhR ligands were dissolved in DMSO and control cells received 0.1% DMSO. Sentence was added in line 135-6.

iv.iv. Figure 2 should be improved. The inclusion of the IFNg effects make it hard to compare effects on IDO1 and IDO2. I suggest to remove the effects of IFNg and adjust the scale to make it simple the comparison between Fig 2A and Fig 2 B. The IFNg results can be included in the legend since making a figure to show only the effects of IFNg is useless with such difference in response (the effects on IDO2 will be to small to be seen.

Answer: The authors agree, and we removed the data for IFNg in figure 2A and B.

iv.v. Figure 5: may not the statistical significance be presented with the p< 0.05)?

Answer: The authors revisited the tools used to analyze IDO2 expression in cancer and confirmed data consistency across tools (including cBioPortal.org). After extensive research on analysis tools used in published literature, we have chosen the TIMER2.0 tool (http://timer.cistrome.org/) and the Breast Cancer Gene-Expression Miner v4.9 tool (http://bcgenex.ico.unicancer.fr/BC-GEM/GEM-Accueil.php?js=1) for the manuscript, each of which has embedded statistical analysis. Newly added Figure 5 uses the TIMER2.0 tool to assess IDO2 expression across cancer types in the TCGA dataset, with normal samples included as available. Results achieving statistical significance are indicated by *, ** or *** (Wilcoxon test). Figure 6 and newly added Figure 7 use the Breast Cancer Gene-Expression Miner v4.9 tool to assess IDO2 expression in breast cancer. This tool uses the Welch’s t-test and when three or more groups are compared, the Dunnet-Tukey-Kramer test. We find that IDO2 expression is significantly increased in most cancer types (Figure 5) and in breast cancer (Figures 5 & 6).

iv.v. A comparison between the different stages can not be done for the different cancer subtypes?

 Answer: The authors revisited the tools used to analyze IDO2 expression in cancer and have concluded that although there are consistent trends towards increased expression of IDO2 in Stage 1 and Stage 2 breast cancer (as compared to normal), these results do not achieve statistical significance. Therefore, we have removed this data from the manuscript.

(v.) Discussion

The discussion section has two distinct parts with a break at line 380. After line 381 it is easy to follow the rational. It is discussed the results obtained in the present study in the MCF-7 cells. The part before line 380 is a more general discussion of previous works of the group. In my view, it would be preferable to include these information after and not before the discussion of  present data. 

 Answer:  Thank you for your feedback. As this is a stylistic preference, we respectfully request to keep the discussion structure as is.

Furthermore, the analysis of breast cancer datasets are not discussed. The absence of this set of results in the discussion and in the aims, make it hard to understand the link between the in vitro studies and the dataset analysis. It seems that they were “artificially” combined with a rational to be studied in parallel. 

Answer: As mentioned above, very little is known regarding IDO2 expression in cancer. Evaluating its expression in human tumor samples provides some insight into its role in cancer.  We found that across different TCGA tumor samples, IDO2 expression is generally increased in cancer compared to normal, which adds to the evidence that IDO2 plays a pro-tumorigenic role in these cancer types. We acknowledge that further functional studies are needed, beyond the scope of this manuscript (Lines 373-376).

(vi)  There is a lack of consistency in the references format. It seems the authors forgot to review this part of the manuscript.

Answer: References have been changed to the correct format for the “Cells”.

Reviewer 2 Report

Sarah et al. investigate the impact of Aryl hydrocarbon Receptor (AhR) activation by endogenous compounds and environmental pollutants on the expression of Indoleamine 2,3-dioxygenase 2 (IDO2) in breast cancer cells. The study found that AhR activation induces IDO2 expression through a xenobiotic response element (XRE) in the promoter region of the ido2 gene. The findings suggest that AhR-mediated expression of IDO2 in breast cancer could contribute to a protumorigenic microenvironment in breast cancer. Overall, the study highlights the potential role of IDO2 in breast cancer and the importance of understanding the mechanisms that regulate its expression.

Although some details need further explanation, I have some questions related to the methods and results. Here are some questions and comments that the authors should think about as they make changes to their work.

1.     Blank controls are absent in many studies. For instance, the author expounded on the statement that "TCDD did not change the expression of IDO2 in MCF-7 AhR knockout cells," based on Figure 1. Unfortunately, we failed to spot any data in Figure 1 with blank controls.  The blank control should be a sample without TCDD treatment. As the information on IDO2 expression in response to TCDD treatment was acquired within that context, it should be a relative expression of the amount. Without TCDD treatment, we are unsure of the exact expression levels of the wt and KO samples. The conclusion mentioned above is thus false. The only result that can be produced is "The expression of IDO2 in MCF-7 AhR knockout cells under TCDD treatment settings indicates a correlated stability." Similarly to this, studies connected to Figure 2 do not include a blank control.

2.     When making horizontal comparisons, it's crucial to ensure that the concentrations of various inducers are uniform. The concentrations of the various inducers changed as the author carried out the experiment shown in Figure 2. What is the basis for the considerations? Moreover, the concentration units are not all the same.

3.     Figure 3b suggests that there may be substantial changes between the WT control group and the AhR-\- control group in IDO2 expression, even though the author did not do statistical analyses on this.

4.     According to Figure 5C, there is a 0.119 P-value between the Luminal and TNBC groups and a 0.06 P-value between the HER2 and TNBC groups. It seems that there are questions about this data. Since Luminal's intergroup variance and mean are plainly smaller and lower than those of the HER2 group, Luminal exhibits a more notable difference than TNBC between HER2 and TNBC in statistical tests. The rules of mathematics are not followed in this. Also, Figure 5's vertical scale is unexpected. Why is the highest limit of TPM values for IDO2 gene expression significantly different from other values in the same dataset?

5.     The criteria of the Altered group and Unaltered group are unclear in light of Figure 6. The definition of this phrase offered by the author is insufficient. The horizontal axis scale identification in Figure 6A is incomplete.

6.     The article mentions Figure 1A in the descriptions in lines 214–215. However, there are no subgraphs in Figure 1. 

Author Response

Reviewer 2

We thank the Reviewer for their careful analysis of our manuscript and comments which have improved our manuscript.

Sarah et al. investigate the impact of Aryl hydrocarbon Receptor (AhR) activation by endogenous compounds and environmental pollutants on the expression of Indoleamine 2,3-dioxygenase 2 (IDO2) in breast cancer cells. The study found that AhR activation induces IDO2 expression through a xenobiotic response element (XRE) in the promoter region of the ido2 gene. The findings suggest that AhR-mediated expression of IDO2 in breast cancer could contribute to a protumorigenic microenvironment in breast cancer. Overall, the study highlights the potential role of IDO2 in breast cancer and the importance of understanding the mechanisms that regulate its expression.

Although some details need further explanation, I have some questions related to the methods and results. Here are some questions and comments that the authors should think about as they make changes to their work.

  1. Blank controls are absent in many studies. For instance, the author expounded on the statement that "TCDD did not change the expression of IDO2 in MCF-7 AhR knockout cells," based on Figure 1. Unfortunately, we failed to spot any data in Figure 1 with blank controls.  The blank control should be a sample without TCDD treatment. As the information on IDO2 expression in response to TCDD treatment was acquired within that context, it should be a relative expression of the amount. Without TCDD treatment, we are unsure of the exact expression levels of the wt and KO samples. The conclusion mentioned above is thus false. The only result that can be produced is "The expression of IDO2 in MCF-7 AhR knockout cells under TCDD treatment settings indicates a correlated stability." Similarly to this, studies connected to Figure 2 do not include a blank control.

Answer: Blank controls and the level of IDO2 in control cells is now included in Figure 1 and 2.

  1. When making horizontal comparisons, it's crucial to ensure that the concentrations of various inducers are uniform. The concentrations of the various inducers changed as the author carried out the experiment shown in Figure 2. What is the basis for the considerations? Moreover, the concentration units are not all the same.

Answer: The concentrations of the AhR ligands and TRAP PM were chosen based on the different binding affinity of the specific AhR ligands as described in line 215.

  1. Figure 3b suggests that there may be substantial changes between the WT control group and the AhR-\- control group in IDO2 expression, even though the author did not do statistical analyses on this.

Answer: Statistics have been added to figure 3b.

  1. According to Figure 5C, there is a 0.119 P-value between the Luminal and TNBC groups and a 0.06 P-value between the HER2 and TNBC groups. It seems that there are questions about this data. Since Luminal's intergroup variance and mean are plainly smaller and lower than those of the HER2 group, Luminal exhibits a more notable difference than TNBC between HER2 and TNBC in statistical tests. The rules of mathematics are not followed in this. Also, Figure 5's vertical scale is unexpected. Why is the highest limit of TPM values for IDO2 gene expression significantly different from other values in the same dataset?

Answer: Thank you for this feedback. In the first version of the manuscript, we used the The University of ALabama at Birmingham CANcer data analysis Portal (https://ualcan.path.uab.edu/index.html).  Based on the Reviewer’s concern, we reached out to UALCAN for insight into their embedded statistical analysis but did not receive a reply. We therefore sought out other data analysis tools which have been widely used in the published literature. We performed the current analyses using cBioPortal (not shown), GEPIA2 (not shown), TIMER2.0 and bc-GenExMiner 4.5 and confirmed that the results we obtained were consistent across the tools we used. We have provided an updated figure (Figure 6) which depicts the differences in IDO2 expression in normal vs. tumor samples (Figure 6A) and in the PAM50 molecular subtypes of breast cancer (Figure 6B). We removed the panel depicting IDO2 expression in breast cancer stages because differences did not achieve statistical significance.

  1. The criteria of the Altered group and Unaltered group are unclear in light of Figure 6. The definition of this phrase offered by the author is insufficient. The horizontal axis scale identification in Figure 6A is incomplete.

Answer: Thank you for this feedback and for prompting further investatigation. We have removed the survival analysis from the manuscript because we found that across different breast cancer datasets, different associations of IDO2 with patient survival can be observed. This will require a more extensive analysis, beyond the scope of this manuscript.

  1. The article mentions Figure 1A in the descriptions in lines 214–215. However, there are no subgraphs in Figure 1. 

Answer: The description has been corrected for figure 1.

Round 2

Reviewer 1 Report

Most of the points raised in the previous evaluation were addressed. The manuscript was very much improved. My congratulations to the authors for their work.

Reviewer 2 Report

he authors responded to all of my concerns carefully. I have no further questions.